# Epigenome-wide association studies identify DNA methylation associated with kidney function

Audrey Y. Chu[1,2], Adrienne Tin[3], Pascal Schlosser [4], Yi-An Ko[5], Chengxiang Qiu[5], Chen Yao[1,2], Roby Joehanes [6,7,1,2], Morgan E. Grams[3], Liming Liang[8], Caroline A. Gluck [5], Chunyu Liu[1,2], Josef Coresh [3], Shih-Jen Hwang[1,2], Daniel Levy[1,2], Eric Boerwinkle[9], James S. Pankow[10], Qiong Yang [11,2], Myriam Fornage[9], Caroline S. Fox[1,2], Katalin Susztak[5] & Anna Köttgen[3,4]

Chronic kidney disease (CKD) is defined by reduced estimated glomerular filtration rate (eGFR). Previous genetic studies have implicated regulatory mechanisms contributing to CKD. Here we present epigenome-wide association studies of eGFR and CKD using whole-blood DNA methylation of 2264 ARIC Study and 2595 Framingham Heart Study participants to identify epigenetic signatures of kidney function. Of 19 CpG sites significantly associated ($P < 1e-07$) with eGFR/CKD and replicated, five also associate with renal fibrosis in biopsies from CKD patients and show concordant DNA methylation changes in kidney cortex. Lead CpGs at *PTPN6/PHB2*, *ANKRD11*, and *TNRC18* map to active enhancers in kidney cortex. At *PTPN6/PHB2* cg19942083, methylation in kidney cortex associates with lower renal *PTPN6* expression, higher eGFR, and less renal fibrosis. The regions containing the 243 eGFR-associated ($P < 1e-05$) CpGs are significantly enriched for transcription factor binding sites of EBF1, EP300, and CEBPB ($P < 5e-6$). Our findings highlight kidney function associated epigenetic variation.

[1] The Population Sciences Branch, Division of Intramural Research, NHLBI, NIH, Bethesda, MD 20892, USA. [2] NHLBI's Framingham Heart Study, Framingham, MA 01702, USA. [3] Department of Epidemiology, Johns Hopkins Bloomberg School of Public Health, Baltimore, MD 21205, USA. [4] Institute of Genetic Epidemiology, Faculty of Medicine and Medical Center—University of Freiburg, 79106 Freiburg, Germany. [5] Renal Electrolyte and Hypertension Division, Department of Medicine, Department of Genetics, University of Pennsylvania, Perelman School of Medicine, Philadelphia, PA 19104, USA. [6] Institute of Aging Research, Hebrew Senior Life, Boston, MA 02131, USA. [7] Department of Medicine, Beth Israel Deaconess Medical Center and Harvard Medical School, Boston, MA 02215, USA. [8] Department of Biostatistics, Harvard University School of Public Health, Boston, MA 02115, USA. [9] Human Genetics Center, University of Texas Health Science Center, Houston, TX 77030, USA. [10] Division of Epidemiology & Community Health, School of Public Health, University of Minnesota, Minneapolis, MN 55454, USA. [11] Department of Biostatistics, Boston University School of Public Health, Boston, MA 02118, USA. Audrey Y. Chu, Adrienne Tin and Pascal Schlosser contributed equally to this work. Correspondence and requests for materials should be addressed to A.K. (email: anna.koettgen@uniklinik-freiburg.de)

Chronic kidney disease (CKD) can progress to end-stage renal disease and is a major contributor to cardiovascular morbidity and mortality[1,2]. The biological mechanisms leading to CKD and its progression are incompletely understood. CKD prevalence increases from <5% in young adults to >40% among those aged ≥70 years[3]. CKD can be defined as the sustained presence of reduced glomerular filtration rate, estimated from serum creatinine concentrations (estimated glomerular filtration rate (eGFR))[4]. A histopathological feature of CKD is renal fibrosis, with tubulo-interstitial fibrosis being a strong marker of CKD progression[5].

Heritability estimates for eGFR and familial aggregation studies of CKD support a substantial heritable component[6–10], of which only a small part is due to classical monogenic diseases. Rather, CKD susceptibility is influenced by DNA sequence variants in many genes, environmental factors, and their interactions. Genome-wide association studies (GWAS) have successfully identified common variants at >60 genetic loci that associate with kidney function and CKD[11–14]. The lead genetic variants for CKD together explain <5% of eGFR variance. Integration of chromatin annotation maps with results of the largest GWAS meta-analysis of eGFR to date supports the importance of altered transcriptional regulation as a mechanism contributing to CKD[14]. Thus, the investigation of epigenetic changes that may influence transcription and associate with eGFR and CKD is of particular interest[15].

DNA methylation is a key regulator of transcription. Epigenome-wide association studies (EWAS) are a cost-efficient, high-throughput, and accurate way to study genome-wide differences in DNA methylation at CpG sites (CpGs) with single-base resolution[16]. Few previously published studies have identified differentially methylated sites from whole blood in association with CKD or its progression[17–19]. These studies were limited by small sample size, cross-sectional design, lack of replication, and/or the presence of comorbidities. A complementary study of DNA methylation assessed from kidney tissues of CKD patients and controls implicated epigenetic regulation of core fibrotic pathways in CKD[20].

To expand on previous work in the field and address limitations of early studies, here we present our study that had the following aims: first, to use EWAS to discover and validate CpGs at which methylation quantified from whole blood is associated with eGFR and the presence and development of CKD among up to 4859 aging adults from population-based studies; second, to characterize validated CpGs for association with fibrosis and gene expression in kidney tissue; and third, to identify common pathways and mechanisms that may link differential DNA methylation to reduced kidney function. We identify and replicate 19 eGFR- and CKD-associated CpGs from whole blood, five of which show concordant and significant changes between DNA methylation quantified from kidney cortex of CKD patients and renal fibrosis. Direction-consistent changes in renal gene expression implicate PTPN6 at the fibrosis-, eGFR-, and CKD-associated CpG site at PTPN6/PHB2. We observe significant enrichment of eGFR-associated CpGs in regions that bind the transcription factors (TFs) EBF1, EP300, and CEBPB. Our findings highlight kidney function-associated epigenetic modifications.

## Results

**Study sample characteristics.** The characteristics of 2264 African American participants from the Atherosclerosis Risk in Communities (ARIC) Study and 2595 European ancestry participants from the Framingham Heart Study (FHS) are summarized in Table 1. Characteristics of both cohorts were comparable to previously published reports[21,22].

**Table 1 Baseline clinical characteristics of the Atherosclerosis Risk in Communities Study (ARIC) and the Framingham Heart Study (FHS). Continuous variables are summarized as mean (SD) unless otherwise noted, and categorical variables as proportion (N)**

| Cohort | ARIC (N = 2264) | FHS (N = 2595) |
|---|---|---|
| Ancestry | African American | European American |
| Age, years | 56.2 (5.8) | 66.3 (8.9) |
| Male | 37.0 (837) | 45.7 (1187) |
| eGFR, mL/min per 1.73 m$^2$ | 106.0 (29.1) | 80.5 (18.8) |
| eGFR <60 mL/min per 1.73 m$^2$ | 3.3 (75) | 11.5 (298) |
| Diabetes | 25.9 (586) | 15.2 (394) |
| Hypertension | 55.9 (1266) | 63.2 (1640) |
| BMI, kg/m$^{2a}$ | 30.1 (6.2) | 28.3 (5.3) |
| LDL-C, mg/dL, median (1st, 3rd quartile)$^a$ | 132.2 (107.8, 159.0) | 103.6 (83.4, 125.0) |
| HbA1c, %$^a$ | 6.3 (1.7) | 5.7 (0.7) |
| High sensitivity C-reactive protein, mg/L, median (1st, 3rd quartile)$^a$ | 3.4 (1.5, 7.1) | 1.5 (0.8, 3.2) |
| Current smoker$^a$ | 23.9 (542) | 8.5 (220) |

$^a$Covariates for sensitivity analysis (ARIC n = 2199; FHS = 2549)

**EWAS of eGFR as well as prevalent and incident CKD.** In the two studies in parallel, we investigated the association of renal traits with DNA methylation at >440,000 cytosines (CpGs) interrogated with the Illumina HumanMethylation450 BeadChip in whole blood. The flowchart of association analysis in ARIC and FHS and a summary of the main results are displayed in Fig. 1. Both studies performed data cleaning and normalization procedures, and related natural log-transformed eGFR, prevalent CKD, and incident CKD (iCKD) to DNA methylation using regression models with residualized methylation β-values as the independent variable (Methods section).

In the ARIC study, methylation levels at 137 CpGs were associated with eGFR at the pre-specified discovery P-value threshold of 1e-5 in multivariable-adjusted analyses (Supplementary Data 1), 11 CpGs were associated with prevalent CKD (Supplementary Data 2), and 29 CpGs were associated with iCKD (Supplementary Data 3). Manhattan plots and QQ plots of association P-values are shown in Supplementary Figs. 1–3. In the FHS study, 53 CpGs were associated with eGFR at the pre-specified significance threshold (1e-5) (Supplementary Data 4), 16 CpGs were associated with prevalent CKD (Supplementary Data 5), and 8 CpGs were associated with iCKD (Supplementary Data 6). Manhattan plots and QQ plots of association P-values are shown in Supplementary Figs. 4–6. The significantly associated CpGs in each study were then reciprocally tested for replication in the other study.

**Independent replication of EWAS results.** Successful replication of an associated CpG was defined as consistent direction of association, a one-sided P-value below a Bonferroni-corrected threshold for the total number of CpGs investigated for each outcome across both cohorts (Methods section) and attaining epigenome-wide significance (P < 1e-7) in a meta-analysis of both studies. These criteria were met by 12 of the eGFR-associated and 3 of the prevalent CKD-associated CpGs discovered in the ARIC study (Tables 2 and 3). Similarly, these criteria were met by 8 of the eGFR-associated and 4 of the prevalent CKD-associated CpGs in FHS (Tables 2 and 3).

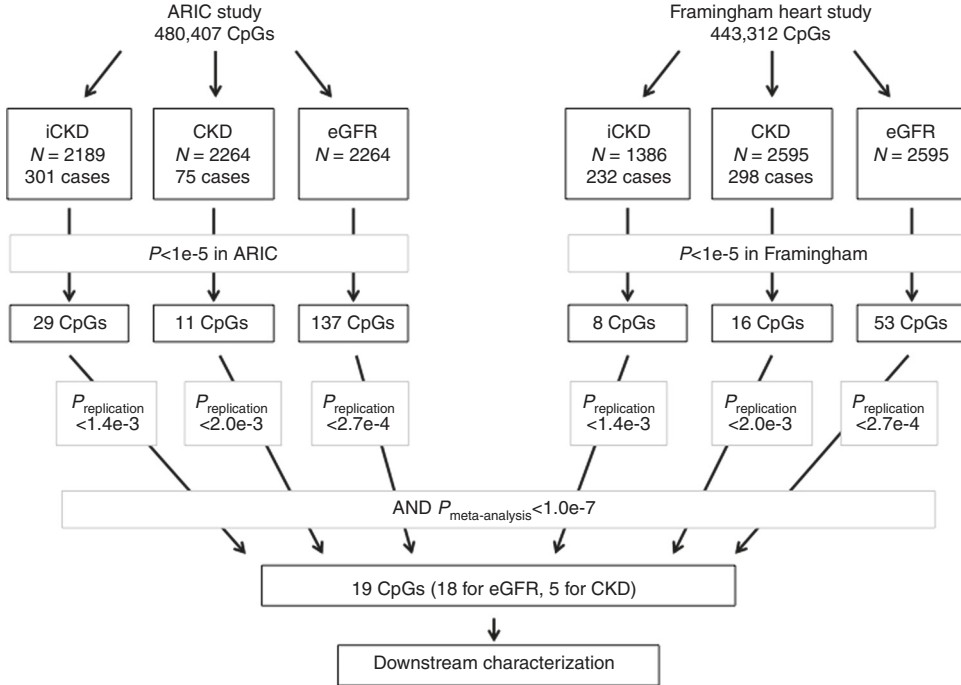

**Fig. 1** Flowchart of the study design and summary of main results. The graph illustrates the number of CpGs that were examined at each step of the analysis and lists the total number of CpGs that were statistically significant at pre-defined thresholds. The 19 replicated CpGs are unique; 4 of the 18 eGFR-associated sites were also associated with CKD

In total, 19 unique CpGs were replicated: 18 CpGs for eGFR (Table 2), and 5 for prevalent CKD (Table 3) of which 4 were also associated with eGFR (cg23597162 in *JAZF1*, and intergenic cg17944885, cg19942083, and cg04036920), and none for iCKD. In the meta-analysis of eGFR-associated sites across both studies, the lowest *P*-value was observed for cg17944885 located between *ZNF788* and *ZNF20* on chromosome 19 ($P = 1.2e-23$). Together, the 18 validated eGFR-associated CpGs explained 1.2% of the variance in eGFR. Mean methylation at these 19 CpGs was similar and highly correlated across the two studies (Pearson's correlation 0.97, Supplementary Data 7), with many sites showing intermediate levels of methylation. Supplementary Data 8 reports details on genes at or near each validated CpGs.

**Sensitivity analyses for replicated CpGs**. To further characterize the 19 replicated CpGs, sensitivity analyses were performed to assess the impact of potential confounders. Associations of the 19 CpGs were robust to adjustment for albuminuria (a marker of kidney damage; Methods section) and remained significant after adjustment. Since five of the 19 CpGs were located in regions previously implicated in EWAS or GWAS for body mass index (BMI), current smoking, and hemoglobin A1c (HbA1C) and LDL-cholesterol (Supplementary Data 8), adjustment for these traits was performed. Associations were generally robust to adjustment (Supplementary Data 9), with little change in beta estimates (change [eGFR] 0.4–21.8% and [CKD] 2.1–8.6%).

The presence of common single-nucleotide polymorphism (SNP) (minor allele frequency (MAF)>0.05) in or within 10 bp of the 19 replicated CpGs was assessed to evaluate whether the presence of SNPs could affect probe binding. One probe, cg22515589, was located near a common SNP, rs4969273. The CpG-eGFR association was minimally changed by adjustment for the SNP (beta-coefficient: 0.035 vs. 0.034), indicating this association is unlikely to be confounded by nearby DNA sequence variation.

We next assessed whether the 19 replicated CpGs that were associated with renal traits cross-sectionally were also associated with iCKD (Methods section). Two CpGs were significantly associated with iCKD after multiple testing correction for 19 tests ($P < 2.6e-3$), cg17944885 on chromosome 19 and cg19942083 between *PTPN6* and *PHB2* on chromosome 12; Supplementary Data 10).

To evaluate whether genetic and epigenetic associations for kidney function localized to the same genomic regions and whether any methylation differences could potentially be explained by correlated genetic variation, we interrogated SNPs in the 1Mb genomic region surrounding the 19 validated CpGs in a GWAS meta-analysis of eGFR conducted in the CKDGen Consortium[14]. A region on chromosome 16 showed significant associations with eGFR (index SNP rs164748, Supplementary Data 11). However, the genotype at the SNP was not significantly associated with methylation in the FHS.

**Previously implicated kidney disease-associated CpGs**. Two prior EWAS of advanced CKD and CKD progression identified 32 CpGs[17,18], but were conducted in studies with <500 participants each. We assessed the association of these 32 previously implicated candidate CpGs with eGFR, prevalent and iCKD in the combined results from ARIC and FHS ($N = 4859$) to evaluate the question whether these associations translate to kidney function traits in population-based studies. Three previously reported sites mapping into the genes *MCM2*, *EXOC3*, and *JARID2* were significantly associated with one or more traits after correction for multiple testing ($P < 1.6e-03$, Supplementary Data 12), and another seven showed nominally significant associations ($P < 0.05$).

**Translation to kidney tissue**. Next, DNA methylation levels at the 19 replicated sites were quantified from the cortical tubule portion of 95 microdissected human kidney samples (Methods

**Table 2 Replicated DNA methylation sites associated with eGFR.** DNA methylation sites associated with eGFR ($P<$1e-5) in ARIC and FHS that were successfully replicated in the other cohort after correction for multiple testing ($P_{Bonferroni}<$2.7e-4) and attained epigenome-wide significance ($P_{meta}<$1e-7) in a fixed effects inverse-variance weighted meta-analysis of ARIC and FHS. The Bonferroni threshold was set at 2.7e-4 = 0.05/182, the number of unique CpG sites brought forward for replication in both ARIC and FHS

| Probe name | Chr | Position (b37) | Gene | ARIC discovery | | | FHS replication | | | Meta-analysis results | | |
|---|---|---|---|---|---|---|---|---|---|---|---|---|
| | | | | $\beta$ | SE | P-value | $\beta$ | SE | P-value(one-sided) | $\beta$ | SE | P-value |
| *DNA methylation sites associated with eGFR that were discovered in ARIC and replicated in FHS* | | | | | | | | | | | | |
| cg11950754 | 1 | 53782077 | LRP8 | −0.037 | 0.007 | 4.2E-07 | −0.024 | 0.006 | 2.1E-05 | −0.029 | 0.005 | 2.0E-10 |
| cg19497511 | 2 | 238609807 | LRRFIP1 | −0.034 | 0.008 | 7.2E-06 | −0.020 | 0.006 | 2.3E-04 | −0.025 | 0.005 | 3.5E-08 |
| cg00501876 | 3 | 39193251 | CSRNP1 | −0.069 | 0.009 | 2.9E-13 | −0.030 | 0.007 | 7.2E-06 | −0.044 | 0.006 | 3.6E-15 |
| cg04460609 | 4 | 16532808 | LDB2 | −0.041 | 0.008 | 1.4E-06 | −0.022 | 0.006 | 1.4E-04 | −0.028 | 0.005 | 7.8E-09 |
| cg09022230 | 7 | 5457225 | TNRC18 | −0.050 | 0.009 | 1.3E-07 | −0.025 | 0.006 | 5.1E-05 | −0.033 | 0.005 | 5.2E-10 |
| cg23597162[a] | 7 | 28102341 | JAZF1 | −0.080 | 0.013 | 3.4E-10 | −0.056 | 0.009 | 3.3E-11 | −0.064 | 0.007 | 2.8E-19 |
| cg02059849 | 8 | 142437898 | PTP4A3 | 0.048 | 0.009 | 8.8E-08 | 0.024 | 0.006 | 1.0E-04 | 0.032 | 0.005 | 7.7E-10 |
| cg10750182 | 10 | 73497514 | C10orf105;CDH23 | −0.060 | 0.012 | 7.3E-07 | −0.033 | 0.008 | 1.6E-05 | −0.041 | 0.007 | 5.0E-10 |
| cg06158227 | 15 | 43662311 | TUBGCP4; ZSCAN29 | −0.042 | 0.008 | 7.8E-08 | −0.020 | 0.005 | 9.9E-05 | −0.027 | 0.004 | 8.5E-10 |
| cg16428517 | 16 | 3317428 | MEFV / ZNF263 | −0.036 | 0.007 | 6.6E-07 | −0.022 | 0.005 | 6.1E-06 | −0.026 | 0.004 | 1.2E-10 |
| cg22515589 | 17 | 79426432 | BAHCC1 | 0.036 | 0.008 | 8.7E-06 | 0.020 | 0.006 | 2.6E-04 | 0.025 | 0.005 | 6.4E-08 |
| cg17944885[a] | 19 | 12225735 | ZNF788/ZNF20 | −0.038 | 0.007 | 1.6E-07 | −0.047 | 0.006 | 1.0E-17 | −0.044 | 0.004 | 1.2E-23 |

| Probe name | Chr | Position (b37) | Gene | FHS discovery | | | ARIC replication | | | Meta-analysis results | | |
|---|---|---|---|---|---|---|---|---|---|---|---|---|
| | | | | $\beta$ | SE | P-value | $\beta$ | SE | P-value (one-sided) | $\beta$ | SE | P-value |
| *DNA Methylation sites associated with eGFR that were discovered in FHS and replicated in ARIC* | | | | | | | | | | | | |
| cg12065228 | 1 | 19652788 | PQLC2 | 0.034 | 0.007 | 2.9E-07 | 0.032 | 0.009 | 1.1E-04 | 0.033 | 0.005 | 2.2E-10 |
| cg23597162[a] | 7 | 28102341 | JAZF1 | −0.056 | 0.009 | 6.6E-11 | −0.080 | 0.013 | 1.7E-10 | −0.064 | 0.007 | 2.8E-19 |
| cg04036920 | 11 | 33562503 | C11orf41 | −0.069 | 0.010 | 1.1E-11 | −0.065 | 0.016 | 3.4E-05 | −0.068 | 0.009 | 2.4E-15 |
| cg19942083 | 12 | 7070562 | PTPN6/PHB2 | 0.030 | 0.006 | 3.7E-06 | 0.038 | 0.009 | 2.0E-05 | 0.033 | 0.005 | 7.2E-10 |
| cg27660627 | 16 | 89461803 | ANKRD11 | −0.050 | 0.010 | 1.0E-06 | −0.056 | 0.015 | 1.3E-04 | −0.052 | 0.008 | 9.9E-10 |
| cg12116137 | 17 | 1576449 | PRPF8 | 0.043 | 0.009 | 5.5E-06 | 0.040 | 0.011 | 1.3E-04 | 0.042 | 0.007 | 5.3E-09 |
| cg00994936 | 19 | 1423902 | DAZAP1 | 0.040 | 0.008 | 3.9E-07 | 0.045 | 0.010 | 1.0E-05 | 0.041 | 0.006 | 3.3E-11 |
| cg17944885[a] | 19 | 12225735 | ZNF788/ZNF20 | −0.047 | 0.006 | 2.0E-17 | −0.038 | 0.007 | 8.0E-08 | −0.044 | 0.004 | 1.2E-23 |

[a]CpG sites with $P<$1e-5 in both ARIC and FHS

**Table 3 Validated DNA methylation sites associated with prevalent CKD.** DNA methylation sites associated with CKD ($P<$1e-5) in ARIC ($N_{cases}=75$, $N_{controls}=2189$) and FHS ($N_{cases}=298$, $N_{controls}=2297$), that were successfully replicated in the other cohort after correction for multiple testing ($P_{Bonferroni}<$2.0e-3) and attained epigenome-wide significance ($P_{meta}<$1e-7) in a combined fixed effects inverse-variance weighted meta-analysis of ARIC and FHS. The Bonferroni threshold was set at 2.0e-3 = 0.05/25, the number of unique CpG sites brought forward for replication in both ARIC and FHS

| Probe name | Chr | Position (b37) | Gene | ARIC discovery | | | FHS replication | | | Meta-analysis results | | |
|---|---|---|---|---|---|---|---|---|---|---|---|---|
| | | | | OR | 95% CI | P-value | OR | 95% CI | P (one-sided) | OR | 95% CI | P-value |
| *DNA methylation sites associated with prevalent CKD that were discovered in ARIC and replicated in FHS* | | | | | | | | | | | | |
| cg23597162 | 7 | 28102341 | JAZF1 | 2.80 | 1.82–4.30 | 2.6E-06 | 1.58 | 1.25–2.00 | 7.8E-05 | 1.81 | 1.47–2.22 | 2.4E-08 |
| cg19942083[a] | 12 | 7070562 | PTPN6/PHB2 | 0.41 | 0.29–0.58 | 3.8E-07 | 0.66 | 0.54–0.79 | 4.6E-06 | 0.59 | 0.50–0.69 | 2.8E-10 |
| cg17944885[a] | 19 | 12225735 | ZNF788/ ZNF20 | 2.34 | 1.79–3.06 | 5.6E-10 | 1.57 | 1.33–1.87 | 1.0E-07 | 1.77 | 1.53–2.04 | 1.2E-14 |

| Probe Name | Chr | Position (b37) | Gene | FHS Discovery | | | ARIC Replication | | | Meta-analysis Results | | |
|---|---|---|---|---|---|---|---|---|---|---|---|---|
| | | | | OR | 95% CI | P-value | OR | 95% CI | P (one-sided) | OR | 95% CI | P-value |
| *DNA methylation sites associated with prevalent CKD that were discovered in FHS and replicated in ARIC* | | | | | | | | | | | | |
| cg19683780 | 8 | 42907576 | FNTA/HOOK3 | 1.65 | 1.35–2.01 | 7.3E-07 | 2.28 | 1.47–3.53 | 1.1E-04 | 1.74 | 1.45–2.08 | 1.7E-09 |
| cg04036920 | 11 | 33562503 | C11orf41 | 1.93 | 1.48–2.52 | 1.4E-06 | 3.07 | 1.77–5.32 | 3.3E-05 | 2.11 | 1.66–2.68 | 1.2E-09 |
| cg19942083[a] | 12 | 7070562 | PTPN6/PHB2 | 0.66 | 0.54–0.79 | 9.2E-06 | 0.41 | 0.29–0.58 | 1.9E-07 | 0.59 | 0.50–0.69 | 2.8E-10 |
| cg17944885[a] | 19 | 12225735 | ZNF788/ ZNF20 | 1.57 | 1.33–1.87 | 2.0E-07 | 2.34 | 1.79–3.06 | 2.8E-10 | 1.77 | 1.53–2.04 | 1.2E-14 |

[a]CpG sites with $P<$1e-5 in both ARIC and FHS

**Table 4 Validated CpGs and their association between methylation in human renal tubular cells and kidney function (eGFR) as well as % renal fibrosis among patients with diabetic kidney disease. The Bonferroni threshold was set at 2.6e-3 = 0.05/19. Significant *P*-values are marked in bold. Human kidney methylation data is based on the data in Ko et al., Genome Biology 2013[20]**

| Probe name | ARIC and FHS meta-analysis results | | | Validation cohort results | | | |
| --- | --- | --- | --- | --- | --- | --- | --- |
| | Blood-based associations | | | Kidney-based associations | | | |
| | eGFR or CKD | | | eGFR | | Fibrosis | |
| | *β*/OR | SE/95% CI | *P*-value | *β* | *P*-value | *β* | *P*-value |
| cg00501876 | −0.044 | 0.006 | 3.6E-15 | −0.0058 | 0.04 | 0.0138 | 8.3E-03 |
| cg00994936 | 0.041 | 0.006 | 3.3E-11 | −0.0055 | 0.30 | 0.0234 | 0.01 |
| cg02059849 | 0.032 | 0.005 | 7.7E-10 | −0.0024 | 0.37 | 0.0060 | 0.25 |
| cg04036920 | −0.068 | 0.009 | 2.4E-15 | −0.0029 | 0.52 | −0.0179 | 0.03 |
| cg04460609 | −0.028 | 0.005 | 7.8E-09 | 0.0017 | 0.62 | −0.0140 | 0.03 |
| cg06158227 | −0.027 | 0.004 | 8.5E-10 | −0.0026 | 0.18 | 0.0079 | 0.02 |
| cg09022230 | −0.033 | 0.005 | 5.2E-10 | −0.0071 | 7.1E-03 | 0.0156 | **1.2E-03** |
| cg10750182 | −0.041 | 0.007 | 5.0E-10 | 0.0025 | 0.31 | −0.0026 | 0.58 |
| cg11950754 | −0.029 | 0.005 | 2.0E-10 | −0.0006 | 0.73 | 0.0057 | 0.10 |
| cg12065228 | 0.033 | 0.005 | 2.2E-10 | 0.0056 | 0.02 | −0.0165 | **1.2E-04** |
| cg12116137 | 0.042 | 0.007 | 5.3E-09 | 0.0109 | 9.2E-03 | −0.0317 | **2.5E-05** |
| cg16428517 | −0.026 | 0.004 | 1.2E-10 | 0.0024 | 0.33 | −0.0053 | 0.24 |
| cg17944885 | −0.044 | 0.004 | 1.2E-23 | 0.0059 | 0.22 | −0.0248 | 4.7E-03 |
| cg19497511 | −0.025 | 0.005 | 3.5E-08 | 0.0003 | 0.86 | 0.0044 | 0.12 |
| cg19683780[a] | 1.74 | 1.45–2.08 | 1.7E-09 | 0.0028 | 0.38 | 0.0007 | 0.91 |
| cg19942083 | 0.033 | 0.005 | 7.2E-10 | 0.0053 | 0.04 | −0.0168 | **2.6E-04** |
| cg22515589 | 0.025 | 0.005 | 6.4E-08 | −0.0016 | 0.45 | 0.0062 | 0.13 |
| cg23597162 | −0.064 | 0.007 | 2.8E-19 | −0.0003 | 0.88 | 0.0040 | 0.36 |
| cg27660627 | −0.052 | 0.008 | 9.9E-10 | −0.0086 | 0.06 | 0.0265 | **1.4E-03** |

Statistical significance defined as *P*<2.6e-3
[a]CpG site associated only with CKD

section) to evaluate whether the CpG associations with kidney traits identified in whole-blood translated to kidney tissue. For each patient from whom a kidney sample was taken, eGFR was calculated and the degree of fibrosis in the renal biopsy was scored by a histopathologist.

Of the 19 replicated CpGs, five showed significant association with % renal fibrosis (*P* < 2.6e-3, Table 4). These CpGs were cg12065228 in *PQLC2*, intergenic cg19942083 near *PTPN6/PHB2*, cg12116137 in *PRPF8*, cg09022230 in *TNRC18*, and cg27660627 in *ANKRD11*. For each of these five CpGs, hypermethylation was associated with higher eGFR and lower percentage of renal fibrosis, consistent with fibrosis as a pathological correlate of CKD and marker of impaired kidney function (Supplementary Figs. 7–10). The direction of association between eGFR and DNA methylation quantified from whole blood was consistent with the one observed from kidney tissue.

Next, chromatin annotation and histone modification maps from the ENCODE Project and from human kidney cortex were used to annotate the regions around the five eGFR-associated CpGs that translated to kidney tissues to investigate their potential functional importance. Figure 2 shows the 200 kb region surrounding cg19942083 on chromosome 12, where the associated index CpG maps downstream of *PTPN6* (encoding protein tyrosine phosphatase non-receptor type 6) and *PHB2* (encoding the mitochondrial protein prohibitin-2). Histone modification marks (H3K27ac and H3K4me1) generated from human whole kidney cortex (Methods section) provided evidence of an active enhancer state surrounding cg19942083. Predicted chromatin states from the 9 common ENCODE cell lines also supported that cg19942083 lies within an enhancer region. Both *PTPN6* and *PHB2*[23–26] have been implicated in experimental models of kidney disease. This, combined with multiple lines of epigenetic evidence, suggests that regulation of one or both of

these genes may be involved in kidney pathogenesis. In addition to the region near *PTPN6/PHB2*, the index CpGs at *ANKRD11* and *TNRC18* also mapped into active enhancers in kidney tissues (Supplementary Figs. 11–14).

Transcriptomic evidence may help to prioritize *PTPN6* or *PHB2* as a candidate for future functional investigations. We therefore tested the association between DNA methylation at cg19942083 and the expression of both genes in kidney tissue. While cg19942083 methylation was not associated with *PHB2* transcript levels in kidney tissue (*P* = 0.87), it was associated with transcript levels of *PTPN6* (*P* = 3.2e-03). Figure 3 shows significant associations for hypermethylation at cg19942083 with lower renal fibrosis (*P* = 1e-07, Fig. 3a), higher eGFR (*P* = 6.4e-06, Fig. 3b), and lower PTPN6 expression (*P* = 3.2e-3, Fig. 3c), suggesting lower renal *PTPN6* expression is associated with a more favorable kidney function profile. In support, higher renal *PTPN6* expression was significantly associated with a greater amount of fibrosis (*P* = 3.3e-5, Fig. 3d). Together, these complementary results suggest that *PTPN6* may be a promising candidate gene for further experimental evaluation and that the focused evaluation of CpGs associated with kidney function from blood-based screens in target tissues can help to prioritize CpGs that may be functionally relevant to disease.

**Transcription factor binding and pathway enrichment analyses.** To identify common features among the kidney function-associated CpGs, we performed transcription factor binding site (TFBS) as well as pathway enrichment analyses. These analyses were based on the 243 CpGs (Supplementary Data 13) that showed suggestive association with eGFR (*P* < 1e-05) in a meta-analysis of ARIC and FHS (Methods section) to maximize statistical power. The CpGs that showed

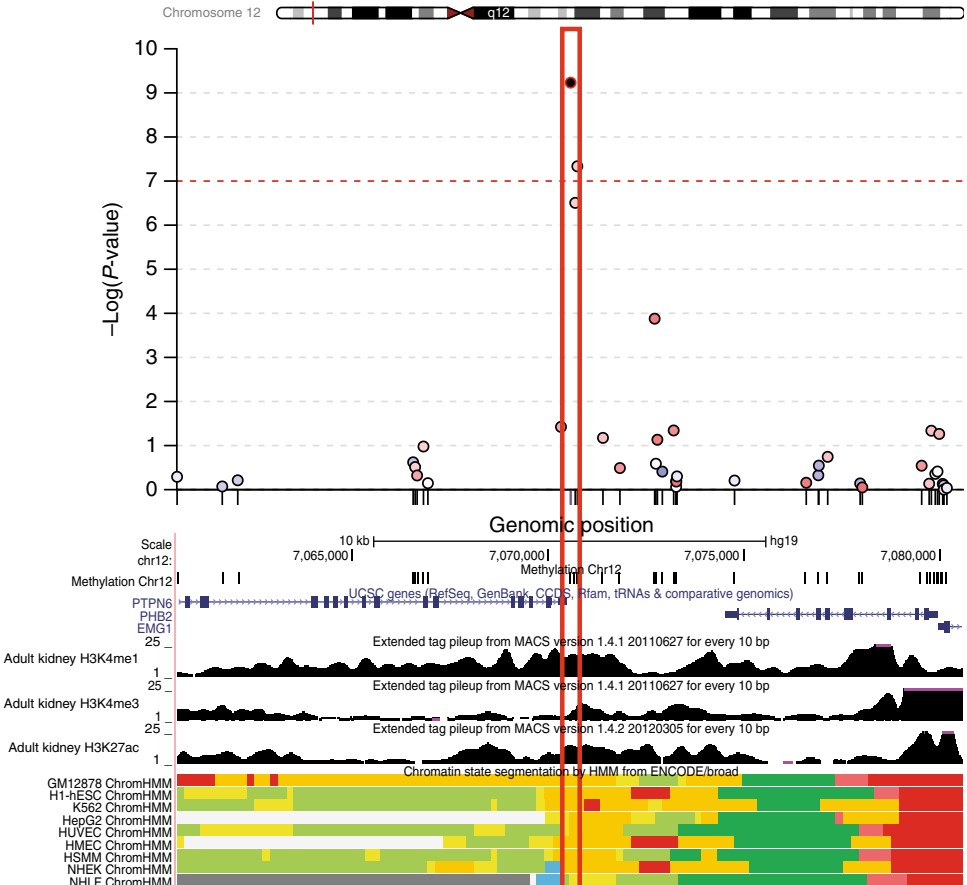

**Fig. 2** Regional association plot and genomic annotation surrounding the index cg19942083 CpG site on chromosome 12p13. Y-axis: -log$_{10}$(P-values) for association between methylation and eGFR from a combined analysis of ARIC and FHS; x-axis: chromosomal position. Each dot represents an evaluated CpG, the color-coding reflects the degree of correlation of methylation values at all other CpGs with that of the index site (red: positive correlation, blue: negative correlation). The index probe cg19942083 is represented as a black dot and included in a red box to facilitate mapping across annotation tracks. Chromatin annotation from various tissues and histone modification marks from kidney cells (black tracks) show that the index probe maps into an active enhancer in kidney tissues. H3K4me1 marks represent poised enhancer elements, H3K4me3 marks represent transcription start sites for actively transcribed genes, and H3K27ac marks represent active enhancer elements. Chromatin annotation track color-code: dark red: active promoter, light red: weak promoter, dark green: transcriptional transition/elongation, light green: weak transcribed, yellow: weak/poised enhancer, orange: strong enhancer, blue: insulator, gray: repressed polycomb, light gray: heterochromatin;low signal;repetitive/CNV. Chromatin annotation tracks were evaluated from 9 different cell lines from the ENCODE Project: B-lymphoblastoid cells (GM12878), embryonic stem cells (H1 hESC), erythrocytic leukemia cells (K562), hepatocellular carcinoma cells (HepG2), umbilical vein endothelial cells (HUVEC), mammary epithelial cells (HMEC), skeletal muscle myoblasts (HSMM), normal epidermal keratinocytes (NHEK), and normal lung fibroblasts (NHLF)

suggestive association with CKD and iCKD (Supplementary Datas 14–15) were not analyzed with respect to TFBSs due to their smaller number (47 and 29 respectively).

First, we evaluated whether eGFR-associated CpG sites preferentially mapped into binding sites of 169 transcription factors (TFs) based on ChIP-seq data from the ENCODE project that were aggregated by consensus calls over 91 human cell types including eight renal tissue based tracks (Methods section). After multiple testing correction for 169 TFs, CEBPB ($P = 3.8e-07$), EBF1 ($P = 1.6e-06$), and EP300 ($P = 1.2e-06$) showed significant enrichment for eGFR-associated CpGs (Fig. 4), and no TF showed depletion. Supplementary Data 16 lists the CpGs driving the association with the binding sites of each of the three TFs.

Second, to assess if CpGs associated with eGFR at $P < 1e-05$ were enriched for annotation to certain pathways or biological processes, gene annotation and pathway analyses were performed as contained in the GO and KEGG databases (Methods section). There were no significantly enriched GO pathways using the results for CKD, eGFR, and iCKD, nor KEGG pathways using the

CKD and iCKD results. In contrast, significant enrichment (FDR<0.01) was identified for 43 KEGG pathways using the association dataset for eGFR. These pathways represented a wide range of processes, especially related to multiple signaling pathways, endocrine or metabolic control, or regulation of cell shape (Supplementary Data 17).

## Discussion

Through EWAS in up to 4859 European ancestry and African American participants of two population-based studies, we identified and replicated DNA methylation level measured in whole blood at 19 CpGs that were associated with eGFR or CKD at epigenome-wide significance. Together, the index CpGs explained 1.2% of the eGFR variance. Five of these CpGs showed significant and direction-consistent association between DNA methylation quantified from human kidney cortex samples and the degree of fibrosis in kidney biopsies, a pathological correlate of CKD. The index CpG at *PTPN6/PHB2* mapped into an active

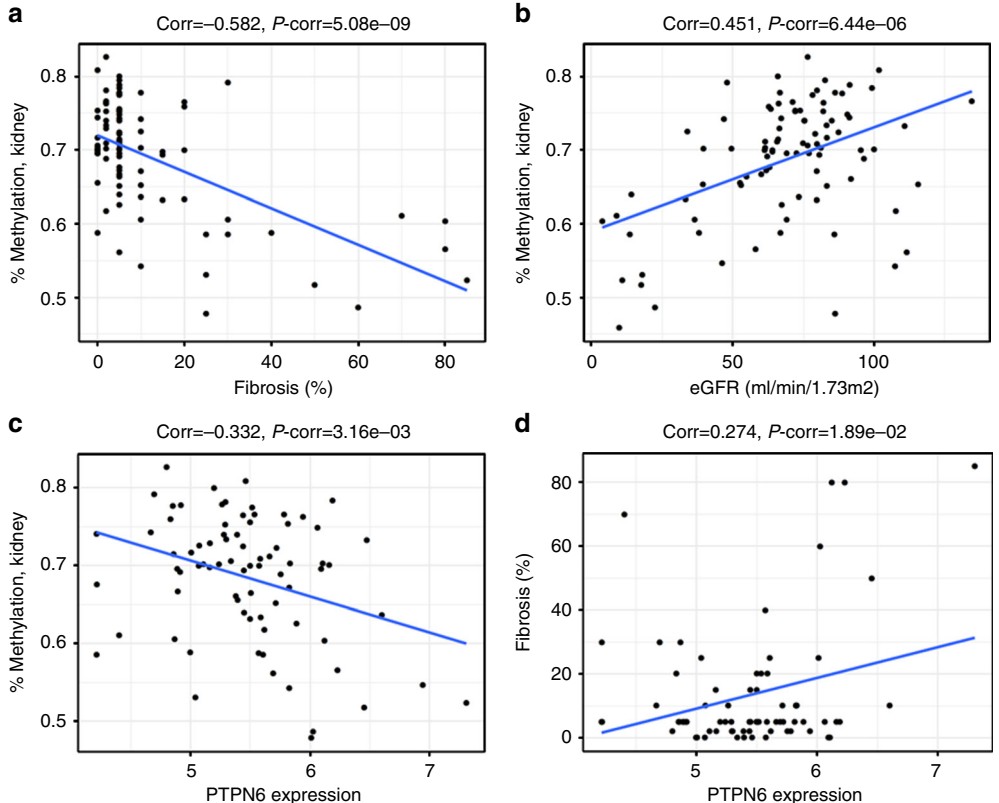

**Fig. 3** Association between DNA methylation of cg19942083 at *PTPN6/PHB2* in kidney tissue and renal fibrosis, eGFR, and gene expression. DNA methylation was quantified from 95 kidney tissue samples. Higher DNA methylation at cg19942083 on chromosome 12 is correlated with lower % fibrosis (*P* = 5.1e-09, **a**) and higher eGFR (*P* = 6.4e-06, **b**) in patients with CKD, indicating less tissue damage and better kidney function. Of the two genes adjacent to the CpG, *PTPN6* and *PHB2*, higher DNA methylation in kidney was associated with lower transcript levels of *PTPN6* (*P* = 3.2e-03, **c**). Supporting the direction of observations A-C, higher *PTPN6* transcript levels were associated with higher % fibrosis (*P* = 1.9e-02, **d**)

enhancer in human kidney cortex cells and was associated with renal *PTPN6* expression, complemented by an association between higher renal *PTPN6* expression and kidney fibrosis. eGFR-associated CpGs were enriched for localization in binding sites of the TFs CEBPB, EBF1, and EP300. These results highlight regions in which DNA methylation is associated with kidney function and CKD and suggest potential pathways of importance for kidney function in health and disease.

Two smaller EWAS of CKD or CKD progression using the HM450K array on whole blood were performed studying DNA methylation of 40 CKD patients with rapid vs. no CKD progression[17], and comparing 255 CKD cases including advanced CKD from type I diabetes and 152 controls[18]. Of 32 candidate sites implicated across these studies, three showed significant association with eGFR or CKD in our study, and 10 showed associations with *P* < 0.05. The lack of association of the other previously implicated CpGs may be related to differences in the severity or causes of CKD in the study populations, the phenotype studied, differences in statistical data analysis, or a combination of the above. Thus, our study confirms some of the previously reported kidney disease associated CpGs including *ANKRD11* at epigenome-wide significance, while discovering and replicating 18 additional loci. Moreover, our study advances knowledge not only because of its larger sample size and replication step, but also because of the examination of the implicated CpGs for association with iCKD over time and for association with DNA methylation, structural changes and gene expression in human kidney tissues.

Of the 19 replicated eGFR- and/or CKD-associated CpGs, five (in or near *PTPN6/PHB2*, *ANKRD11*, *TNRC18*, *PQLC2*, and *PRPF8*) also showed a significant association between DNA methylation from human kidney cortex samples and the degree of fibrosis in the corresponding kidney biopsies. All five sites showed the expected direction of association: sites for which hypermethylation was associated with better kidney function showed association with lower kidney fibrosis and vice versa, consistent with the association between lower eGFR and increased fibrosis, a pathological readout of CKD and a predictor of CKD progression. Some of these loci are further supported by additional analyses, including association with renal transcript levels of *PTPN6*. These findings suggest that the study of DNA methylation in whole blood can provide insight with potential translation to additional tissues such as the kidney.

Existing biological evidence supports the relevance of some of the identified associated CpGs for kidney function and disease. *PTPN6* encodes protein tyrosine phosphatase, non-receptor type 6, also known as Src homology-2 domain-containing phosphatase-1 (SHP-1). Consistent with our observations, increased renal SHP-1 expression has been implicated in several experimental studies of kidney disease and vascular complications in the setting of diabetes[25,27]. Another biologically plausible finding from our study is the enrichment of eGFR-associated CpGs at binding sites of the TFs EBF1. A recent study reported a previously unnoticed renal phenotype in Ebf1 null mice, including glomerular maturation defects during development, resulting

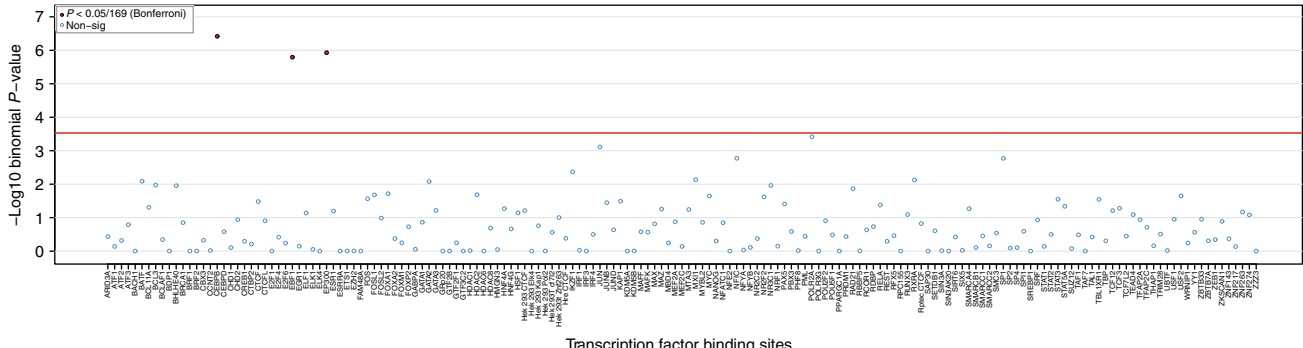

**Fig. 4** Enrichment analysis of CpGs significantly associated with eGFR for mapping into regions containing specific transcription factor binding sites. *Y*-axis: -log₁₀(*P*-value) from a binomial test comparing the expected and observed numbers of significant CpGs that map into the binding site regions for a given transcription factor. *X*-axis: 169 evaluated transcription factors, listed in alphabetical order. Across 169 evaluated transcription factors, CpGs associated with eGFR at *P*<1e-05 in the meta-analysis of ARIC and FHS show significantly enriched mapping into regions that contain binding sites for transcription factors CEBPB, EBF1, and EP300 (red color). Enrichment testing was carried out using permutation with matching for genomic localization when sampling from the background. Statistical significance was set at *P*<3e-04 (red line), corresponding to a Bonferroni correction for 169 transcription factors

in proteinuria and reduced eGFR[28]. These findings are consistent with the notion that murine Ebf1 regulates target genes important for kidney development and function, and the authors hypothesized that alterations in promoter/enhancer occupancy by Ebf1 may be a mechanism related to kidney damage. EWAS in human populations cannot provide direct mechanistic evidence, but can use rigorous statistical methods to highlight targets of human relevance for future integration with experimental evidence.

Changes in DNA methylation can precede changes in kidney function or CKD development, or occur as their consequence. When we investigated the association between the 19 CpGs associated with cross-sectional kidney traits and iCKD, only two of the CpGs were associated with iCKD, and none-at epigenome-wide significance. These results support the notion that associations between differential methylation in whole blood with eGFR and CKD may occur as consequences rather than the cause of kidney disease, a phenomenon which was recently observed for obesity-associated CpGs[26]. Alternatively, statistical power may have been insufficient to detect associations at the stringent epigenome-wide significance level. Therefore, while the CpG sites associated with cross sectional traits may highlight genes and pathways associated with kidney function and disease, the CpG sites by themselves are unlikely to be useful biomarkers to predict iCKD in population-based studies.

Strengths of our study include the investigation of clinically relevant phenotypes, its large sample size, ethnic diversity, independent replication of findings, and rigorous analytical methods. In addition, a variety of resources were used in the characterization of our findings including DNA methylation, structural changes, and gene expression in human target tissues, as well as enrichment testing that appropriately matched for genomic localization. Replication in samples of different ancestries shows that our results are generalizable; the trans-ethnic replication design of our study likely resulted in a conservative estimate of associated CpGs because it does not focus on ethnicity-specific findings. Multi-ancestry replication has been successfully carried out in other recent EWAS with the reported replicated sites showing little trans-ethnic heterogeneity[22,26]. Future large-scale EWAS meta-analyses of kidney function will be able to investigate the presence of ethnicity-specific kidney function associated CpGs.

Despite the strengths of this study, there are some limitations that warrant mention. DNA methylation for the discovery EWAS was quantified from whole blood, a cell type mixture. We addressed this issue by adjustment for imputed white blood cell

composition and by independent replication of our findings. These procedures reduce concern about confounding but do not completely abolish it. However, our study and other recent publications support that trait-associated differentially methylated sites in blood can show similar associations in the target tissue[21,26,29,30]. This suggests that DNA methylation in blood can serve as a proxy of methylation in other tissues, which is supported by our finding that a subset of the reported sites were also associated with DNA methylation in kidney tissue. In addition, it is known that specific forms of CKD have an immunological component, raising the possibility that blood cells may represent additional extra-renal target tissues relevant to kidney function and disease. Another limitation, inherent to the use of the 450K array, is that it covers only ~2% of total human DNA methylation and therefore limits the discovery potential of such screens.

In conclusion, this well-powered EWAS of kidney function and CKD in 4859 participants from the general population identified differential DNA methylation significantly and reproducibly associated with kidney function and CKD as well as with the clinical endpoint renal fibrosis. Genomic annotation and enrichment tests implicate *PTPN6* and target genes of CEBPB, EBF1, and EP300 as promising candidates for future experimental studies to illuminate the underlying gene regulatory mechanisms linking differential DNA methylation to kidney function in health and disease.

## Methods

**Study population.** The ARIC Study is an ongoing prospective cohort study in four US communities[31]. A total of 15,792 participants aged 45–64 years were recruited from Forsyth County, North Carolina; Jackson, Mississippi (African Americans only); suburban Minneapolis, Minnesota; and Washington County, Maryland between 1987 and 1989 (Visit 1). Four follow-up examinations (Visits 2–5) were conducted. Measures of DNA methylation in peripheral blood leukocyte samples were available for 2879 African Americans study participants from Visit 2 (1990–92) and Visit 3 (1993–95). Of the 2796 samples available after DNA methylation quality control process described below, we further excluded 379 samples with DNA methylation quantified at Visit 3 due to the lack of serum creatinine measures at Visit 3. Of the remaining 2417 samples with measures of DNA methylation at Visit 2, the baseline visit for this analysis, samples were excluded due to missing data values at Visit 2: serum creatinine (*N* = 39), hypertension and diabetes (*N* = 56), and imputed white blood cell counts (*N* = 58). This resulted in a sample size of 2264 for the cross-sectional analysis of eGFR and CKD at Visit 2. Further, for the analysis of iCKD, we excluded 75 participants with eGFR <60 mL/min per 1.73 m² at Visit 2, resulting in 2189 participants for the prospective analyses. Written informed consent was obtained from all study participants. The relevant Institutional Review Boards approved the study protocols.

The original FHS cohort was established in 1948 and consisted of 5209 women and men of European ancestry from Framingham, Massachusetts[32]. In 1971, the Framingham Offspring Cohort was established and consisted of 5124 women and

men who were children or the spouses of children from the original cohort[33]. Offspring participants attending the eighth (2005–08) examination cycle with valid DNA methylation data, the baseline visit for this analysis, were eligible ($N = 2759$), of which 2595 had available eGFR values. Of these, 1386 were free of baseline CKD and returned at the next visit and had eGFR assessed again. All participants provided written informed consent. The Institutional Review Boards of the Boston University Medical Center approved the study.

**DNA methylation quantification and quality control.** For the quantification of DNA methylation in the ARIC study, genomic DNA was extracted from peripheral blood leukocyte samples using the Gentra Puregene Blood Kit (Qiagen; Valencia, CA, USA) and underwent bisulfite conversion using the EZ-96 DNA Methylation Kit (Zymo Research Corporation; Irvine, CA, USA) according to the manufacturers' instructions. Levels of DNA methylation at 485,577 probe sites were quantified using the Illumina Infinium HumanMethylation450K Beadchip array (HM450K). Illumina GenomeStudio Methylation module 1.9.0 was used to extract the intensity value of each site and perform background correction. The methylation level at each site was represented as a beta ($\beta$) value of the fluorescent intensity ratio ranging from 0 (nonmethylated) to 1 (completely methylated). The wateRmelon package was used to conduct quality filtering[34]. The Beta Mixture Quantile Dilation (BMIQ) method was used to adjust the beta values of type 2 design probes on the array to the statistical distribution characteristic of type 1 probes[35]. We excluded probe sites with detection $P$-value >0.01, beadcount <3 in ≥5% of the sample and missing in ≥1% of the sample, resulting in a total of 480,407 sites for analysis. We further excluded samples ($n = 83$) having ≥1% of the probe sites with detection $P$-value>0.01 or missing, SNP mismatch between HM450K array and microarray data (Affymetric 6.0, Exome Chip, IBC chip, Metabochip), or outliers in multi-dimensional scaling analysis. After quality control, there were a total of 2796 samples and 480,407 CpG sites.

In FHS, genomic DNA was isolated from buffy coat preparations obtained from whole-blood samples (Gentra Puregene Blood Kit—Qiagen, Venlo, Netherlands) and underwent bisulfite conversion (EZ DNA Methylation kit—Zymo Research, Irvine, CA)[36]. DNA methylation was assayed using the HM450K. The methylated probe intensity and total probe intensities were extracted using the Illumina Genome Studio (version 2011.1) with the methylation module (version 1.9.0). Methylation level at each site was represented as a beta ($\beta$) value, the ratio of the methylated signal (M) to the sum of the methylated and unmethylated signal (U), $\beta = M/(M+U+100)$. Normalization of raw DNA methylation values was performed within laboratory batches using the DASEN methodology implemented in the wateRmelon package[34]. Samples with missing rate >1% at $p < 0.01$, poor SNP matching to the 65 SNP control probe locations, and outliers by multi-dimensional scaling techniques were excluded. Probes with missing rate of >20% at $p < 0.01$ were also excluded. Data from laboratory batches were pooled leaving a total of 443,312 probes for subsequent analysis. Additional information on DNA methylation, normalization, and quality control is available in Asbeykian et al[36].

**Outcome definition.** In both the ARIC study and FHS, eGFR was calculated using the four-variable Modification of Diet in Renal Disease (MDRD) equation[37], to be consistent with most published GWAS of kidney function. CKD was defined as eGFR <60 mL/min per 1.73 m².

In the ARIC study, serum creatinine values was measured by the modified Jaffe assay and calibrated to standard reference materials[38]. Participants were followed from Visit 2 (1990–92) until 30 September 2013. iCKD was defined as eGFR ≥60 mL/min per 1.73 m² at Visit 2 and eGFR <60 mL/min per 1.73 m² at Visit 4 (1996–98) or Visit 5 (2011–13).

In the FHS, serum creatinine was measured using the modified Jaffe method from fasting blood samples collected during the eighth and ninth exams. Serum creatinine measures can vary widely across different laboratories. Therefore, we calibrated serum creatinine values through a two-step process: (1) calibration of National Health and Nutrition Examination Survey III (NHANES III) creatinine values to the Cleveland Clinic Laboratory resulting in a correction factor of 0.23 mg/dl, and (2) alignment of mean serum creatinine values from the Framingham Offspring Study by sex-specific age groups (20–39, 40–59, 60–69, and ≥70 years) with the corresponding corrected NHANES III age- and sex-specific mean[39]. Baseline for defining prevalence CKD was the eighth examination. iCKD was defined as the presence of an eGFR <60 mL/min per 1.73 m² at the ninth examination (follow-up) among participants free with eGFR ≥ 60 mL/min per 1.73 m² at the eighth examination. Only those participants free of prevalent CKD were included in the iCKD analysis.

**Covariate assessment.** In the ARIC study, prevalent diabetes was defined as fasting plasma glucose ≥126 mg/dl, non-fasting plasma glucose ≥200 mg/dl, treatment for diabetes, or self-report of a diabetes diagnosis. Prevalent hypertension was defined as systolic blood pressure ≥140 mm Hg, diastolic blood pressure ≥90 mm Hg, or treatment for hypertension. Current smoking was based on self-report. Plasma HbA1c was measured using high performance liquid chromatography. High sensitivity C-reactive protein was measured in serum using a latex-particle enhanced immunoturbidimetric assay kit (Roche Diagnostics, Indianapolis, IN) and read on the Roche Modular P800 Chemistry analyzer (Roche Diagnostics).

In the FHS, diabetes was defined as plasma fasting glucose ≥126 mg/dL or use of oral hypoglycemic agents or insulin. Hypertension was defined as baseline systolic blood pressure >140 mmHg, diastolic blood pressure >90 mmHg, or use of antihypertensive medication. Albuminuria was defined as urinary albumin-to-creatinine ratio (UACR) ≥30. HbA1C was measured at baseline using whole blood on a Roche Hitachi 911 machine (hemolysis followed by a turbidimetric immunoassay). Low density lipoprotein cholesterol (LDL-C) was calculated using the Friedewald equation (total cholesterol—high density lipoprotein cholesterol—triglyderides/5 if triglycerides<400 mg/dL). BMI (kg/m²) was calculated using weight and height measurements. Current smoking was defined as smoking ≥1 cigarette/day in the past year.

**Statistical analysis.** DNA methylation beta values can be skewed. To avoid undue influence from outliers, we performed inverse normal transformation of the beta values for all analyses. Because of the bimodal distribution of the beta values for chromosome X, we performed sex-stratified analyses of sites on the X chromosome. Imputed cell count for neutrophils, lymphocytes, monocytes, and eosinophils was generated based on the Houseman algorithm in both studies[40].

In the ARIC study, to adjust for white blood cell subtype composition and batch effects, we used linear mixed effect models with chip number as the random effect and plate, chip row, and imputed cell counts of neutrophils, lymphocytes, monocytes, and eosinophils as fixed effects to generate residuals of the inverse normal transformed beta values for association analysis with kidney outcomes. To evaluate the association between DNA methylation and eGFR, we used linear regression with natural log-transformed eGFR as the outcome. For the association between DNA methylation and prevalent CKD, we used logistic regression with prevalent CKD as the outcome. For the association between DNA methylation and iCKD, we used Cox proportional hazards regression with iCKD as the outcome. In the primary analyses, covariates for both outcomes were age, sex, study site, prevalent diabetes, and hypertension, as well as 10 genetic principal components generated using genotypes from the Exome Chip array. In addition to these covariates, the analysis for iCKD further adjusted for eGFR at Visit 2. For all analyses, the predictor of interest were the residuals of the inverse normal transformed beta values adjusted for imputed white blood cell count and batch effect. All analyses were performed using R (3.1.0). The mixed effect model analysis was performed using lmer in the lme4 package.

In the FHS, the inverse-quantile normalized values of the methylation $\beta$-values were residualized adjusting for laboratory batch, plate, column, row, and Houseman imputed cell counts[40,41]. To assess the association between DNA methylation and kidney function, regression models were run using normalized and residualized methylation $\beta$-values as the independent variable and natural log-transformed eGFR, prevalent CKD and iCKD as the dependent variables. Two models were assessed: model 1 adjusted for age, sex, diabetes status, and hypertension, and model 2 adjusted for model 1 covariates and albuminuria. Models analyzing iCKD also included baseline eGFR. To account for family relatedness, linear mixed models (lmekin R package) were used to analyze linear outcomes and generalized estimating equations (geepack R package) were used to analyze binary outcomes and (CKD and iCKD). All models were adjusted for laboratory batch (fixed effect), and family structure (random effect).

**Replication and concordance.** We oriented our workflow of discovery, replication, and assessment of epigenome-wide significance in the combined analysis on practices commonly applied in the field of GWAS[11,42]. For all CpGs with association $P<1e-5$ from each trait in either ARIC or FHS, replication of the association was sought for the same trait in the other cohort (i.e. all CpGs with $P<1e-5$ for association with eGFR in ARIC would be examined in FHS, and vice versa). The one-sided Bonferroni-corrected $P$-value threshold for each trait was: $P_{eGFR}<2.7e-4$ (0.05/182 [number of sites tested across both cohorts]); $P_{CKD}<2.0e-3$ (0.05/25); $P_{iCKD}<1.4-3$ (0.05/37). In addition a fixed-effects meta-analysis using inverse-variance weights combining results from both cohorts was performed on the 430,136 shared autosomal CpG sites. This analysis was performed using R version 3.3.0 with the packages metafor[43] and qqman[44]. Furthermore, we performed a meta-analysis using Stouffers method, which is based on a sample-size weighted mean of the z-scores[45]. The results agreed with the fixed-effects meta-analysis and further supported our finding because Stouffers method is more robust to differences in analytical approaches between studies. The following criteria were used to determine replication of CpGs: (1) $P < 1e-5$ in ARIC or FHS, (2) one-sided $P<$Bonferroni correction as indicated above and consistent direction of association, and (3) $P < 1e-7$ ([~0.05/430,136 CpGs]—Bonferroni correction for the total number of probes examined) in the fixed effects meta-analysis.

Directional concordance of all CpGs identified in each cohort at $P < 1e-5$ was examined in the respective other study. For the 125 eGFR-associated and the 11 CKD-associated sites discovered in ARIC and available in FHS, 101 and 7, respectively, showed the same direction of association in FHS ($P_{binomial} = 2.0e-12$ and 0.55, respectively). For sites discovered in FHS and available in ARIC, 48 of the 53 eGFR-associated and 15 of the 16 CKD-associated CpGs showed the same direction of association in ARIC ($P_{binomial} = 7.1e-10$ and 5.2e-4, respectively). Thus, although only a subset of CpGs passed the stringent multiple testing correction required for replication, there was significant directional consistency of the

association between DNA methylation and kidney function across the two studies beyond chance expectation.

**Sensitivity analysis**. For replicated CpGs in or near genes that were previously reported to be associated with type 2 diabetes, LDL-C, BMI, and smoking, we conducted additional sensitivity analyses by adding HbA1c, LDL-C, BMI, and current smoking to the primary analysis model in ARIC and FHS, respectively.

**Targeted investigations of eGFR probes in iCKD**. The association of the 19 eGFR- or CKD-associated and validated CpGs with iCKD was evaluated after correcting for the number of evaluated sites, and statistical significance was defined as $P < 2.6e-3$ (0.05/19).

**Evaluation of eGFR-associated SNPs near validated CpG sites**. To assess whether methylation at the associated CpGs occurred in genomic regions that contain known genetic variants associated with eGFR, we examined the 1 Mb region surrounding each of the 18 replicated CpGs in summary statistics from meta-analyses of GWAS of eGFR from the CKDGen Consortium (up to 133,413 individuals of European ancestry)[14]. Significant association between SNPs and eGFR was defined as $P < 4.07e-6$, corresponding to a Bonferroni correction for 12,292 independent SNPs of MAF>0.01 in the EUR population of the 1000 Genomes Project. Independent SNPs were identified using plink (version 1.90b2), with options—indeppairwise 50 5 0.2 and—maf 0.01.

**Previously reported kidney function or CKD-associated CpGs**. Thirty-two CpGs previously reported as associated with CKD[18] or CKD progression[17] were evaluated for association with the three investigated traits in the meta-analysis results from ARIC and FHS. Statistical significance was defined as $P < 1.56e-03$ (0.05/32).

**Secondary analyses**. The existence of SNPs in HM450K probes can influence measures of DNA methylation[46]. The list of SNPs that could potentially affect methylation levels was obtained from Illumina (http://support.illumina.com/content/dam/illumina-support/documents/downloads/productfiles/humanmethylation450/humanmethylation450_dbsnp137.snpupdate.table.v2.sorted.txt), and the imputed dosage of these SNPs (reference panel 1000 Genomes Phase I March 2012) was used as an additional covariate in the ARIC study to evaluate the association between DNA methylation and the outcomes for the validated sites.

The coMET R package[47] was used to create Fig. 2 and Supplementary Figs. 11–14.

The percentage of phenotypic variance explained by the 18 replicated CpGs associated with eGFR was estimated as $\Sigma_{i=1}^{18} R_i^2$ with $R_i^2 = \beta_i^2 \mathrm{var}(\mathrm{CpG}_i)/\mathrm{var}(\ln(\mathrm{eGFRcrea}))$, where $\beta_i$ is the estimated association of probe $i$ on $\ln(\mathrm{eGFRcrea})$ and $\mathrm{var}(\ln(\mathrm{eGFRcrea}))$ was calculated by pooled variance from the ARIC study and FHS.

**TFBS enrichment analyses**. Prior to enrichment testing, we evaluated whether the genomic localization of associated CpGs differed from the background. Indeed, the distribution of the distance to the nearest transcription start site as well as the genomic localization annotation differed clearly for eGFR-associated and background CpGs highlighting the importance of matching for localization when conducting enrichment tests (Supplementary Fig. 15). Enrichment testing was therefore carried out using permutation matching for genomic localization when sampling from the background.

Enrichment was assessed using eForge[48] to account for the differences of kidney function-associated sites and background sites with respect to gene- and CpG island-region localization by calculation of an empirical $P$-value in a resampling based approach. The 243 CpGs (Supplementary Data 13) that were associated with eGFR at <1e-05 in the meta-analysis were used as input, and 10,000 resampling runs, an active proximity filter and a Bonferroni correction for 169 factors were used as eForge parameters.

**GO and pathway enrichment analysis**. Enrichment of GO terms and KEGG pathways was performed in R version 3.3.0 with the missMethyl package[49] taking into account the number of CpG sites per gene and correcting for multiple testing using the methods of Benjamini and Hochberg[50].

**Kidney tissue cohort and DNA methylation quantification**. DNA was extracted from 94 microdissected human kidney tubule epithelial samples from routine surgical nephrectomies. Samples were de-identified and the corresponding clinical information was collected by an individual not involved in the research protocol. The study was approved by the Institutional Review Boards of the Albert Einstein College of Medicine Montefiore Medical Center (IRB#2002-202) and the University of Pennsylvania. Histological analysis was performed by an expert pathologist (IRB#815796). Dissected tissue was homogenized and genomic DNA extracted with the DNAeasy kit (Qiagen, Hilden, Germany). Control samples were defined by eGFR greater than 60 ml/min per 1.73 m$^2$, absence of significant

proteinuria, and less than 10% fibrosis on histology. Samples with significant hematuria or other signs of glomerulonephritis (HIV, hepatitis, or lupus) were excluded, as well as one sample with missing clinical data. Genomic DNA (200 ng) was purified using the DNeasy Blood and Tissue Kit (Qiagen, Hilden, Germany) following the manufacturer's protocol. Purified DNA quality and concentration were assessed with NanoDrop ND-1000 (Thermo Fisher Scientific, Waltham, MA, USA) and by Quant-iT PicoGreen dsDNA Assay Kit (Life Technologies, Carlsbad, CA, USA) prior to bisulfite conversion.

DNA methylation was measured using the Infinium HumanMethylation450 Beadchip (Illumina, San Diego, CA), with 485,000 CpG sites across the genome examined. Most pre-process and quality control steps were conducted by Minfi package (v1.19.14)[41]. We corrected the raw signal intensities according to the background intensity and then removed probes with a $P$-value >0.01 in any samples. We further removed probes targeting SNPs and probes that contained a SNP within the probe or within 1 base pair to the probe (MAF>0.01 according to dbSNP137). The process yielded 418,329 probes for analysis. One subject was excluded after principal component analysis showed they were not within ±3 standard deviations from the mean of a PC1, PC2, and PC3. The proportion of methylation at each CpG site was taken as the beta value, which was calculated from the intensities of the methylated (m) and unmethylated (u) probes [$\beta = m/(m+u+100)$]. Beta-mixture quantile normalization was performed using RnBeads package (v0.99.19)[51] to adjust for type 2 bias. In addition, the bisulfite conversion efficiency was estimated by the bisulfite conversion control probes according to the Illumina guidelines. The annotation data were acquired from Bioconductor package IlluminaHumanMethylation450kanno.ilmn12.hg19 (v0.2.1).

**Statistical analyses of kidney DNA methylation and fibrosis**. Multiple linear regression was used to assess the relationship between methylation and either eGFR or percent renal fibrosis at baseline. Beta value was transformed to M ($M = \log_2[\mathrm{beta}/(1\text{-beta})]$), and M was taken as the dependent variable while eGFR or fibrosis were the independent variables; covariates included age, sex, race, DM type, HTN type, batch, and conversion efficiency.

**Data availability**. TFBS chromatin immunoprecipitation-sequencing data (161 TFBS in 91 human cell types) was acquired from ENCODE project TFBS clusters, which included 690 datasets that passed quality assessment through the ENCODE March 2012 data freeze (http://hgdownload.cse.ucsc.edu/goldenpath/hg19/encodeDCC/wgEncodeRegTfbsClustered/). The eight individual renal tracks, which were part of the clustered file, were accessed at the ENCODE Uniform Peaks page (http://genome.ucsc.edu/cgi-bin/hgTrackUi?db=hg19&g=wgEncodeAwgTfbsUniform). The GEO accession numbers for the three histone modification tracks from adult kidney were GSM670025 (H3K4me1), GSM621648 (H3K4me3), and GSM1112806 (H3K27ac). The summary statistics from the meta-analysis of the ARIC and FHS studies have been submitted to dbGaP accession phs000930 and can be requested from there. Additional data and programming code that support the findings of this study are available from the authors upon request.

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

## Acknowledgements

We thank the staff and participants of the ARIC study for their important contributions. The ARIC study is carried out as a collaborative study supported by the National Heart, Lung, and Blood Institute (NHLBI) contracts (HHSN268201100005C, HHSN268201100006C, HHSN268201100007C, HHSN268201100008C, HHSN268201100009C, HHSN268201100010C, HHSN268201100011C, and HHSN268201100012C). We thank the staff and participants of the ARIC study for their important contributions. Funding support for "Building on GWAS for NHLBI-diseases: the U.S. CHARGE consortium" was provided by the NIH through the American Recovery and Reinvestment Act of 2009 (ARRA) (5RC2HL102419). The work of P.S. and A.K. was supported by the CRC 992 Initiative and by a Heisenberg Professorship (KO 3598/3-1 to A.K.) of the German Research Foundation (DFG). The FHS is funded by National Institutes of Health contract N01-HC-25195. The laboratory work for this investigation was funded by the Division of Intramural Research, National Heart, Lung, and Blood Institute, National Institutes of Health, Bethesda, MD, and the NIH Director's Challenge Award (PI: D. Levy). The analytical component of this project was funded by the Division of Intramural Research, National Heart, Lung, and Blood Institute, and the Center for Information Technology, National Institutes of Health, Bethesda, MD. Dr. Liang's was supported by P30 DK46200 and partially supported by P30 DK46200. Work in the Susztak lab was supported by NIH DP3 DK108220 and NIH R01 DK087635. The views expressed in this manuscript are those of the authors and do not necessarily represent the views of the National Heart, Lung, and Blood Institute; the National Institutes of Health; or the U.S. Department of Health and Human Services.

## Author contributions

A.Y.C., A.T., P.S., C.S.F., and A.K. contributed to study design. M.E.G., C.A.G., J.C., D.L., E.B., J.S.P., M.F., C.S.F., K.S., and A.K. were involved in participant recruitment, sample collection, or methylation assessment. A.Y.C., A.T., P.S., Y.-A.K., C.Q., C.Y., R.J., L.L., C.A.G., C.L., S.-J.W., Q.Y., and A.K. contributed to statistical methods and analysis.

A.Y.C., A.T., P.S., Y.K., C.Q. and A.K. drafted the manuscript. M.E.G., C.A.G., C.S.F., and K.S. performed critical review of the manuscript.

## Additional information

**Competing interests:** C.S.F. and A.Y.C. are currently employed by Merck Research Laboratories. The remaining authors declare no competing financial interests.

