## [Peer Review File · Nature Communications]

Reviewers' comments:

Reviewer #2 (Remarks to the Author):

This is a very difficult paper to review because it is both very professionally written and unconvincing. I was not persuaded that the main results will stand the test of time, yet it is hard to identify any specific weaknesses.

Things that make me feel uneasy are,

(a) Using two ethnically different populations for cross-replication. There is plenty of evidence in the literature of ethnic differences in methylation and even more evidence of the importance of batch effects in sample collection and processing. Also FHS patients are older and have much worse kidney function.

(b) Methylation may be linked to causes of kidney function and/or be a result of poor kidney function. The lack of understanding of the basic mechanisms makes it difficult to design appropriate studies and it makes it difficult to interpret the results.

(c) This study only replicates 3 or 32 findings from two previous EWAS of kidney function. The authors do not comment on their power to replicate but given the large sample sizes it must be high. I take this to be a consequence of the nature of methylation rather than a criticism of any single study.

(d) The rather arbitrary cut-off used to control the analysis. $1e-5$ in an adjusted analysis then one-sided Bonferroni plus pooled $1e-7$. These choices are not unreasonable but there is no rationale provided and we do not know whether the analysts investigated other thresholds before choosing these. $1e-5$ is described as pre-specified but the status of the adjustments and replication criteria is unclear.

(e) The lack of a false discovery analysis of the findings seems a strange omission. One wonders what effect dependence and slight inflation in the p-values would have on the FDR.

Having stated my concerns it is only fair to say that the authors have produced a well-written account of their analysis and the methods that they have used are not dissimilar to those used in many other papers in genetic epidemiology.

Reviewer #3 (Remarks to the Author):

The authors carried out a meta-analysis with 2,264 African American participants from Atherosclerosis Risk in Communities (ARIC) study and 2,595 European ancestry participants from Framingham Heart Study (FHS) to identify 14 methylation markers associated with estimated glomerular filtration rate (eGFR), 1 associated with chronic kidney disease (CKD), 4 associated with both traits in an epigenome-wide association approach. Among the 19 validated methylation markers, two were associated with incident (iCKD). Moreover, they characterized those methylation markers in kidney fibrosis tissue and also with gene expression. Finally, the authors reported common pathways and mechanisms that could relate methylation to reduced kidney function.

The present work is to date the study which uses the largest sample size to assess the association between DNA methylation and eGFR, CKD and iCKD. Moreover, they deeply characterize those sites showing interesting results that could be useful for future works. However, several concerns have to be addressed.

Comments:

1. Page 4 line 95: The authors wrote the PubMed ID instead of the references.

2. The authors should revise the following numbers in the manuscript:

- 2.1. Page 22 line 471: The authors reported 133 CpGs associated with eGFR, instead of the 137 which are presented in page 5 line 105, figure 1 and Supplementary table 1A.
 - 2.2. Page 5 line 110: The authors reported 57 CpGs whereas in the figure 1, the Supplementary Table 2A and methods section the authors reported 53 CpGs.
 - 2.3. Figure 1: The p-values reported are different that those presented in the methods section (page 22 lines 458-459).
 - 2.4. Page 6 lines 125-126: The authors reported 19 validated CpGs; 18 were associated with eGFR and 5 with prevalent CKD of which 4 were also associated with eGFR. But, in the figure 1 they presented 19 CpGs (18 associated with eGFR and 1 with CKD).
3. Page 14 lines 287-293: Could the authors explain why they think is more likely that the association between methylation markers and eGFR and CDK is the consequence rather than the cause of kidney disease?
4. The study population for ARIC study is clear enough, but for the FHS is not completely clear. In the page 17 line 348 the participants with available DNA methylation data were 2,759, but the tables and the results reported 2,595 participants. Why in the analyses with iCKD trait only 1,386 individuals were used?
5. Page 23 line 492- page 24 line 498 and page 25 line 518-523: The authors explain twice: " A target evaluation of the 1Mb genomic region surrounding each of the 19 replicated CpGs in summary statistics from a meta-analysis of GWAS for eGFR conducted in the CKDGen Consortium".
6. Could the authors clarify that the visit 2 and exam 8 are the baseline visit for ARIC and FHS, respectively?
7. There are medications that directly or indirectly could modify the eGFR. In the present manuscript, the author did not adjust the models for medication. Could the authors explain why they think it is not necessary to adjust for medication?

We thank you for evaluating our manuscript entitled "Epigenome-wide Association Studies Identify DNA Methylation Associated with Kidney Function" (NCOMMS-17-03142) and for the helpful and constructive comments. We have now addressed these comments through additional analyses and by providing more information, and are pleased to submit a substantially revised version of the manuscript. These major revisions are summarized in the point-by-point response below, including where and how we have modified the manuscript.

Reviewer #2 (Remarks to the Author):

1. Using two ethnically different populations for cross-replication. There is plenty of evidence in the literature of ethnic differences in methylation and even more evidence of the importance of batch effects in sample collection and processing. Also FHS patients are older and have much worse kidney function.

Response: We agree with the Reviewer that these topics require attention.

The feasibility of trans-ethnic epigenome-wide association studies (EWAS) is illustrated not only by our analyses but also other recent publications: in an EWAS of obesity, Wahl *et al* systematically compared the evidence for heterogeneity of BMI-associated CpGs across different ethnicities. As illustrated in Suppl. Table 3 of their publication, they found significant evidence for trans-ethnic heterogeneity at only seven out of 187 replicated CpGs.¹ Similar to our study, Ligthart *et al* also examined a European ancestry discovery study sample, followed by replication among African Americans.² They reported that the effect estimates for replicated sites were highly correlated between European and African-American study samples ($r=0.97$) and that Cochran's Q statistics displayed homogeneity for > 95% of replicated loci. These studies also support the presence of trait-associated CpGs that translate across ethnicities. The examination of different ethnicities, as done in our study, can therefore be considered a strength because it highlights kidney function-associated CpGs despite differences in the underlying genetic architecture and other characteristics such as age. Nevertheless, population-specific differentially methylated CpGs likely also exist. Since our study design focused on the identification of trait-associated CpGs across ethnicities, it represents a conservative estimate of kidney function associated CpGs because it does not highlight population-specific differentially methylated sites. We have included a discussion of this aspect in our manuscript on page 15, paragraph 1.

To minimize the influence of potential batch effects on sample collection and processing, several measures have been taken. Both the ARIC study and the FHS follow highly standardized sample collection and processing procedures that include detailed instruction for blood draws, sample processing and storage procedures as well as DNA extraction. At the data processing stage, each study applied appropriate normalization methods to address potential effects of laboratory batches (see Methods section, page 19, paragraph 2). At the data analysis stage, statistical models accounted for laboratory batch, chip number, plate, and chip row and column, as applicable, in order

to further minimize potential batch effects (see Methods, page 22, paragraphs 2 and 3, page 23, paragraph 1).

Finally, by only focusing on replicated CpGs across studies, we minimize the chances of highlighting potential artifacts arising from study-specific batch effects, because it is unlikely that the same batch effects occur in both studies and affect the same set of CpGs. Similarly, focusing only on sites that show evidence for association in both studies (i.e. replicated sites) leads to the identification of sites that are robust to differences across studies, such as differences in mean age.

2. Methylation may be linked to causes of kidney function and/or be a result of poor kidney function. The lack of understanding of the basic mechanisms makes it difficult to design appropriate studies and it makes it difficult to interpret the results.

Response: Because of the dynamic nature of DNA methylation and because humans cannot be subjected to the same experimental settings as model organisms or cells, disentangling cause and effect is a challenge of any human epigenome-wide association study.

To account for potential confounding by cause of kidney dysfunction, we adjusted our analyses for diabetes and hypertension, the most common causes of kidney dysfunction in study populations such as ours. In addition, we examined whether altered DNA methylation precedes the development of chronic kidney disease (CKD) by evaluating the outcome incident, i.e. new-onset CKD over time. We did not identify any CpGs that were significantly associated with incident CKD, which does not support that differential methylation at the examined sites causes the development of new-onset CKD. This is in agreement with a recent large EWAS of obesity, that found statistical evidence that changes in DNA methylation occur as a consequence of obesity rather than its cause,¹ thereby highlighting epigenetic mechanisms that occur as a response to a disease. Because of the relatively modest number of incident CKD cases in our study, however, the question of changes in methylation preceding the onset of CKD should be re-assessed in future, larger EWAS meta-analyses.

Importantly, our study identifies significant associations between DNA methylation at specific genomic sites and eGFR that translate to human kidney tissue and an association with a clinical endpoint, kidney fibrosis. This is quite novel and our study can therefore be considered a starting point for the design of studies investigating the molecular mechanisms of these associations in experimental models, as pointed out by the Reviewer. Studies that start with human relevance and subsequently investigate underlying mechanisms are therefore complementary to studies that start with a detailed mechanistic understanding in model systems but often have not yet established human relevance. We hope that these two study types together will yield novel insights into epigenetic mechanisms of human complex diseases.

Prompted by the Reviewer's comment and in addition to the response above, we tried to think of additional analytical ways to investigate the question of cause and effect. One such analysis builds on the concept of Mendelian Randomization.³ DNA methylation at a given CpG can be influenced by nearby DNA sequence variants.^{4,5} Thus, DNA methylation-associated SNPs in *cis* (meQTLs) can serve as a proxy for life-long genetic predisposition to differential methylation at this site. Assuming that such meQTLs only affect eGFR through their effect on DNA methylation, an observed association between a meQTL SNP and eGFR suggests that changes in DNA methylation may subsequently influence kidney function, rather than *vice versa*.

Therefore, as a first step, we searched for meQTLs for each of the 53 eGFR-associated CpGs discovered in the European ancestry FHS population at $p < 1e-5$. A meQTL was defined as a SNP within 1 Mb of each CpG and associated the DNA methylation levels at the CpG at $p < 5e-08$, the conventional genome-wide significance threshold. In a second step, we confirmed the assumption that these meQTLs were not associated with eGFR after controlling for DNA methylation at these CpGs. As a third step, we evaluated the association of each identified meQTL with eGFR. To maximize statistical power for this third step, we investigated the association between each meQTL SNP and eGFR in results from a large-scale meta-analysis of genome-wide association studies of eGFR in the CKDGen Consortium⁶ that integrates data from 110,517 individuals of European ancestry.

Significant meQTLs were identified for 34 out of 53 CpGs, resulting in a statistical significance threshold of $p < 1.5e-03$ ($0.05/34$) for the tests of association between meQTL and eGFR (third step). The mean proportion of DNA methylation variance explained by these CpGs was 8.0% (range 1-46%). Given the significance threshold of $1.5e-03$, the proportion of DNA methylation variance explained by the meQTLs, and the CKDGen sample size, we had at least 80% power to detect associations between the meQTLs and eGFR across the range of observed CpG-effect sizes in our EWAS.⁷ Despite adequate statistical power, only one out of 34 meQTLs showed an association with eGFR in the CKDGen data ($p = 1.2e-03$). This suggests that differential DNA methylation at these sites does not systematically result in changes in kidney function, which is in line with the lack of association we observed between DNA methylation and incident CKD. Both of these analyses, therefore, do not support that changes in DNA methylation are causative for changes in kidney function. We note, however, that meQTLs were not identified for all eGFR-associated CpGs. We also did not have adequate statistical power to test whether genetic proxies for reduced eGFR are associated with differential DNA methylation, because summary statistics from very large EWAS meta-analyses are not yet available. A formal investigation of this question should therefore be carried out through bi-directional MR analyses in future, larger studies, which we are currently initiating.

3. This study only replicates 3 or 32 findings from two previous EWAS of kidney function. The authors do not comment on their power to replicate but given the large sample sizes it must be high. I take this to be a consequence of the nature of methylation rather than a criticism of any single study.

Response: We agree that differences in study design and/or environmental exposures may contribute to the identification of different associated CpGs across studies. The two previous studies differed in their design in that Wing *et al* studied a decline in kidney function over time among 40 patients with already existing CKD, and Smyth and colleagues compared DNA methylation profiles of 255 CKD patients (44% of whom suffered from advanced CKD attributed to type I diabetes) and 152 controls (74% with type 1 diabetes). We have therefore modified the manuscript to clearly state that the purpose of the lookup was to assess whether DNAm sites previously identified in the setting of advanced CKD are also detected in population-based studies of kidney function, and to avoid the term “replication” because the assessed phenotypes differ (page 8, paragraph 2, page 12, paragraph 2).

4. The rather arbitrary cut-off used to control the analysis. $1e-5$ in an adjusted analysis then one-sided Bonferroni plus pooled $1e-7$. These choices are not unreasonable but there is no rationale provided and we do not know whether the analysts investigated other thresholds before choosing these. $1e-5$ is described as pre-specified but the status of the adjustments and replication criteria is unclear.

Response: The significance threshold of $1e-07$ to indicate epigenome-wide significance (corresponding to correcting for the number of CpGs on the 450K array) is established in the literature. Two-stage EWAS meta-analyses (discovery followed by replication and meta-analysis of discovery and replication) are still relatively recent, and criteria to select CpGs for replication are not yet as firmly established as in genome-wide association studies. We have therefore oriented our workflow on the decade-long experiences in the latter field. Here, the use of a suggestive but not genome-wide significance cutoff to identify candidate variants in discovery, followed by replication testing and the assessment of genome-wide statistical significance in the combined discovery and replication samples is common practice for which multiple examples exist.^{8,9}

The procedures in previous two-stage EWAS publications are consistent with our approach in that they also evaluate evidence for association of candidate CpGs separately in the replication samples and assess epigenome-wide significance in the combined discovery and replication data.^{1,10} We expect that, similar to genome-wide association studies, standard replication workflows will be established more firmly as two-stage EWAS meta-analyses become more commonplace. As suggested by the Reviewer, we have included the rationale behind the selection of our workflow in the manuscript (page 23, paragraph 2).

5. The lack of a false discovery analysis of the findings seems a strange omission. One wonders what effect dependence and slight inflation in the p-values would have on the FDR.

Response: As previous large-scale EWAS such as ours had only presented p-values and not false discovery analyses, we had decided *a priori* to not conduct false discovery analyses. To address the Reviewer's concern, we have now performed these analyses, using the Benjamini-Hochberg FDR. The results of these analyses are summarized for the Reviewer in **Reviewer Table 1**, below.

All five CKD-associated sites (four of which were also associated with eGFR) and all 18 eGFR-associated sites that we replicated in our approach were below a 5% FDR cutoff in the meta-analysis of both studies. While our approach led to replication testing of 182 sites for eGFR, 25 for CKD and 37 for iCKD, using an FDR cutoff of 5% at the discovery stage would have resulted in replication testing of one site for CKD, 218 for eGFR, and none for iCKD. Had we chosen the FDR to assess statistical significance at the discovery stage, we would likely have chosen a less conservative FDR cutoff such as 10%. This cutoff would have resulted in replication testing of 2, 502, and 9 sites for CKD, eGFR, and iCKD, respectively. Importantly, all replicated sites highlighted in our manuscript are among these sites. Thus, the approach chosen in our manuscript to correct for multiple testing, a Bonferroni-corrected p-value of 1e-07, is robust to the FDR method.

Reviewer Table 1: Number of kidney-function associated CpGs at 5% FDR in the individual studies and their meta-analysis. Numbers in parenthesis refer to the replicated CpGs highlighted in our manuscript.

Trait	No. of sig. sites in ARIC	No. of sig. sites in FHS	No of sig. sites in meta-analysis (no. of replicated CpGs with FDR <5% / all replicated CpGs)
CKD	1	0	28 (5/5)
eGFR	187	34	491 (18/18)
iCKD	1	0	0 (0/0)

Reviewer #3 (Remarks to the Author):

Comments:

1. Page 4 line 95: The authors wrote the PubMed ID instead of the references.

Response: Thank you for noticing this. We have included the references.

2. The authors should revise the following numbers in the manuscript:

2.1. Page 22 line 471: The authors reported 133 CpGs associated with eGFR, instead of the 137 which are presented in page 5 line 105, figure 1 and Supplementary table 1A.

Response: We appreciate the thorough reading of our manuscript. The inconsistency was introduced by the fact that not all sites discovered in ARIC were also available for replication testing in the Framingham study. We have revised the numbers to reflect this on page 24, paragraph 2. The number of sites with consistent direction in both studies is unchanged.

2.2. Page 5 line 110: The authors reported 57 CpGs whereas in the figure 1, the Supplementary Table 2A and methods section the authors reported 53 CpGs.

Response: Thank you, we have clarified that 53 is the correct number.

2.3. Figure 1: The p-values reported are different that those presented in the methods section (page 22 lines 458-459).

Response: The p-values in Figure 1 have been adjusted to match the correct ones in the text.

2.4. Page 6 lines 125-126: The authors reported 19 validated CpGs; 18 were associated with eGFR and 5 with prevalent CKD of which 4 were also associated with eGFR. But, in the figure 1 they presented 19 CpGs (18 associated with eGFR and 1 with CKD).

Response: This seeming discrepancy results from the fact that Figure 1 only lists unique CpGs and the four sites associated with both traits are listed as eGFR sites. We have clarified this in the Figure 1 caption.

3. Page 14 lines 287-293: Could the authors explain why they think is more likely that the association between methylation markers and eGFR and CDK is the consequence rather than the cause of kidney disease?

Response: Our explanation in the manuscript is based on the fact that we did not identify any CpG sites significantly associated with new-onset CKD over the course of time (iCKD), including the ones that are significantly associated with kidney function cross-sectionally. In response to this comment as well as to a comment by Reviewer 2, we have now undertaken additional effort to address this point analytically. These analyses are summarized in the response to comment 2 by Reviewer 2, above, and are in agreement with our initial statement.

4. The study population for ARIC study is clear enough, but for the FHS is not completely clear. In the page 17 line 348 the participants with available DNA methylation data were 2,759, but the tables

and the results reported 2,595 participants. Why in the analyses with iCKD trait only 1,386 individuals were used?

Response: In the FHS study, 2,759 study participants were assessed for methylation, but only 2,595 of these had baseline eGFR measurements available. Of these, 1,386 were free of baseline CKD and returned at the next visit and had eGFR assessed again. We have now clarified this in the Methods section (page 18, paragraph 1).

5. Page 23 line 492- page 24 line 498 and page 25 line 518-523: The authors explain twice: “ A target evaluation of the 1Mb genomic region surrounding each of the 19 replicated CpGs in summary statistics from a meta-analysis of GWAS for eGFR conducted in the CKDGen Consortium”.

Response: Thank you for picking this up. We have removed one of the statements.

6. Could the authors clarify that the visit 2 and exam 8 are the baseline visit for ARIC and FHS, respectively?

Response: This is correct. We have now stated this explicitly (page 17, paragraph 1 and page 18, paragraph 1).

7. There are medications that directly or indirectly could modify the eGFR. In the present manuscript, the author did not adjust the models for medication. Could the authors explain why they think it is not necessary to adjust for medication?

Response: We agree that some medications can modify eGFR. This may be especially relevant in patient studies or clinical trials in which many patients use the same medication. We would expect potential confounding only in a scenario in which a specific medication affected both kidney function as well as DNA methylation. As a precaution, however, we have incorporated adjustment for commonly prescribed medication classes in the general population, anti-diabetic and anti-hypertensive medications, into our definitions of diabetes and hypertension that were included as covariates in the statistical analysis. Thus, adjustment for the intake of these medications is already inherent. The incorporation of medication intake information into the definition of the covariates diabetes and hypertension is included on page 20, paragraph 3.

References:

1. Wahl S, Drong A, Lehne B, Loh M, Scott WR, Kunze S, et al. Epigenome-wide association study of body mass index, and the adverse outcomes of adiposity. *Nature*. 2017;541:81-86
2. Ligthart S, Marzi C, Aslibekyan S, Mendelson MM, Conneely KN, Tanaka T, et al. DNA methylation signatures of chronic low-grade inflammation are associated with complex diseases. *Genome Biol*. 2016;17:255
3. Davey Smith G, Hemani G. Mendelian randomization: genetic anchors for causal inference in epidemiological studies. *Hum Mol Genet*. 2014;23:R89-98
4. Chen L, Ge B, Casale FP, Vasquez L, Kwan T, Garrido-Martin D, et al. Genetic Drivers of Epigenetic and Transcriptional Variation in Human Immune Cells. *Cell*. 2016;167:1398-1414 e1324
5. McClay JL, Shabalin AA, Dozmorov MG, Adkins DE, Kumar G, Nerella S, et al. High density methylation QTL analysis in human blood via next-generation sequencing of the methylated genomic DNA fraction. *Genome Biol*. 2015;16:291
6. Gorski M, van der Most PJ, Teumer A, Chu AY, Li M, Mijatovic V, et al. 1000 Genomes-based meta-analysis identifies 10 novel loci for kidney function. *Sci Rep*. 2017;7:45040
7. Brion MJ, Shakhbazov K, Visscher PM. Calculating statistical power in Mendelian randomization studies. *Int J Epidemiol*. 2013;42:1497-1501
8. Kottgen A, Pattaro C, Boger CA, Fuchsberger C, Olden M, Glazer NL, et al. New loci associated with kidney function and chronic kidney disease. *Nat Genet*. 2010;42:376-384
9. Liu C, Kraja AT, Smith JA, Brody JA, Franceschini N, Bis JC, et al. Meta-analysis identifies common and rare variants influencing blood pressure and overlapping with metabolic trait loci. *Nat Genet*. 2016;48:1162-1170
10. Chambers JC, Loh M, Lehne B, Drong A, Kriebel J, Motta V, et al. Epigenome-wide association of DNA methylation markers in peripheral blood from Indian Asians and Europeans with incident type 2 diabetes: a nested case-control study. *Lancet Diabetes Endocrinol*. 2015;3:526-534

REVIEWERS' COMMENTS:

Reviewer #2 (Remarks to the Author):

I should like to thank the authors for their detailed and well-presented responses to my concerns.

The idea of using bi-directional MR is particularly interesting to me. We tried this approach with a much smaller dataset and a different disease outcome but the results were not clear. Perhaps the effects work in both directions.

I remain concerned by the low level of replication but I am happy to accept that my concerns are over issues of opinion rather than them being grounds for scientific criticism and so I have no further comments to make.

Reviewer #3 (Remarks to the Author):

The authors have adressed all my comments.